# Injectable hydrogel electrodes as conduction highways to restore native pacing

Gabriel J. Rodriguez-Rivera[1], Allison Post[2], Mathews John[2], Skylar Buchan[2], Drew Bernard[2], Mehdi Razavi ☉ [2,3] ✉ & Elizabeth Cosgriff-Hernandez ☉ [4] ✉

There is an urgent clinical need for a treatment regimen that addresses the underlying pathophysiology of ventricular arrhythmias, the leading cause of sudden cardiac death. The current report describes the design of an injectable hydrogel electrode and successful deployment in a pig model with access far more refined than any current pacing modalities allow. In addition to successful cardiac capture and pacing, analysis of surface ECG tracings and three-dimensional electroanatomic mapping revealed a QRS morphology comparable to native sinus rhythm, strongly suggesting the hydrogel electrode captures the deep septal bundle branches and Purkinje fibers. In an ablation model, electroanatomic mapping data demonstrated that the activation wavefront from the hydrogel reaches the mid-myocardium and endocardium much earlier than current single-point pacing modalities. Such uniform activation of broad swaths of tissue enables an opportunity to minimize the delayed myocardial conduction of heterogeneous tissue that underpins re-entry. Collectively, these studies demonstrate the feasibility of a new pacing modality that most closely resembles native conduction with the potential to eliminate lethal re-entrant arrhythmias and provide painless defibrillation.

Ventricular arrhythmias are the leading cause of sudden cardiac death in the United States[1]. Current pacing modalities are insufficient to prevent or treat these scarred or otherwise diseased myocardium leads to slowed electrical conduction of cardiac tissue. The attendant spatial and temporal heterogeneities cause wave break, re-entry, and chaotic cardiac activity and fibrillation[2–5]. Current management relies on a combination of antiarrhythmic drugs—all with high toxicity profiles—and/or ablation and destruction of tissues adjacent or in the vicinity of the diseased regions. Antiarrhythmic drugs work by further slowing conduction velocity in an effort towards complete elimination of such conduction[6]. This renders them pro-arrhythmic for exactly the same reason: decreased conduction's requisite role in the initiation and sustenance of re-entry[7]. Although widely adopted, ablative strategies have a high failure rate with recurrent arrhythmias in 18–40% of cases[8]. Of paramount importance, none of these technologies correct

the underpinning mechanism of re-entry: delayed conduction. For many patients, the only option is an implantable cardiac defibrillator that uses high energy shocks to extinguish re-entry circuits. These far exceed the pain threshold and have a well-documented negative impact on quality of life, including post-traumatic stress disorder and depression[9]. A preventative treatment regimen addressing the underlying pathophysiology of re-entry remains elusive.

In this work, we developed an injectable hydrogel electrode that fills the epicardial coronary veins and tributaries, converting them into flexible electrodes that can finally reach heretofore inaccessible mid-myocardium. This approach takes advantage of minimally invasive catheter delivery and standard pacemaker technologies to integrate into clinical workflow and ensure adoption. In addition to increased access to the mid-myocardium, this hydrogel electrode enables simultaneous pacing from multiple sites along the length of the

[1]McKetta Department of Chemical Engineering, The University of Texas at Austin, Austin, TX 787212, USA. [2]Electrophysiology Clinical Research and Innovations, Texas Heart Institute, Houston, TX 77030, USA. [3]Division of Cardiology, Department of Medicine, Baylor College of Medicine, Houston, TX 77030, USA. [4]Department of Biomedical Engineering, The University of Texas at Austin, Austin, TX 78712, USA. ✉e-mail: mehdirazavi1@gmail.com; cosgriff.hernandez@utexas.edu

electrode rather than from a single point stimulus. We hypothesize that the resultant planar wave propagation from the length of the electrode will stimulate wide areas of ventricular tissue that would have otherwise been subject to delayed activation. This normalizes and eliminates the regions of delayed activation that are the underpinning of re-entry. To this end, we developed a hydrogel system that has the requisite conductivity, biostability, and rapid in situ cure to be delivered via a transvenous catheter. First, a polyethylene glycol-based macromer with improved hydrolytic stability and durability was combined with redox chemistry to provide rapid in situ cure without an external stimulus. Ionic species added to the hydrogel precursor solutions were used to confer conductivity above target myocardium values. Intravital delivery and successful pacing via the hydrogel electrode was tested in a porcine model and electroanatomical mapping was used to assess conduction in a porcine cardiac ablation model. Finally, we defined the subacute safety profile of this technique in a porcine model using a combination of cardiac function, cardiac enzymes, and histology. Collectively, this work demonstrates for the first time the ability to confer direct electrical stimulation of the native and scarred mid-myocardium using an injectable hydrogel electrode as a pacing modality.

## Results

### Injectable hydrogels with rapid in situ cure and ionic conductivity

There are no current hydrogels that can provide the multifaceted needs of a cardiac electrode with endovascular delivery to the mid-myocardium: injectability, conductivity, and biostability. Poly(ethylene glycol) (PEG) hydrogels are a popular choice based on their established biocompatibility and highly tunable, soft tissue-like properties; however, traditional acrylate-derivatized PEG hydrogels are susceptible to slow degradation in vivo due to hydrolysis. To generate a PEG-based hydrogel suitable for long-term implantable applications, we synthesized a hydrogel chemistry that combines biostability with flexibility and durability. Polyether urethane diacrylamide (PEUDAm) contains urethane and amide groups that are resistant to hydrolysis at physiological conditions, Fig. 1. The synthetic route is described in Supplementary Fig. 1 and successful conversion (>95%) was confirmed with $H^1$ NMR (Supplementary Figs. 2–4). In addition to the biostable PEG macromer, we synthesized N-acryloyl glycinamide (NAGA), a small molecule crosslinker with bidentate hydrogen bonding[10]. The proposed design also requires rapid in situ cure of the hydrogel without external stimuli (e.g UV). Our lab has previously shown tunable gelation using the redox pair ammonium persulfate (APS) and iron

gluconate (IG)[11]. A double-barrel syringe with mixing head was used to deliver the precursor solution as an analog for future delivery from a dual lumen catheter, Fig. 1. Iterative testing was used to identify a target cure rate less than 1 minute with full network formation in less than 2 minutes, which corresponded to an APS concentrations ≥0.75 mM. The ionic PEUDAm hydrogels rely on salts dissolved in the precursor solution to serve as charge carriers to generate a current when connected to a power source. As a basic proof-of-principle, Fig. 1 demonstrates that a current could be generated by connecting the ionic hydrogel to a battery and a light-emitting diode. In order to activate the myocardium beyond a single point of contact in vivo, the conductivity of the hydrogel must be above the myocardial conductivity that ranges from 0.1 to 6.0 mS/cm[12–14]. The conductivity as measured by the impedance in the plateau regions was 13.5 ± 0.8 mS/cm for the ionic gel as compared to <0.1 mS/cm for the water gel control.

### Successful deployment and safety of injectable hydrogel electrode in a porcine model

To simulate clinical deployment in a beating and perfused heart, precursor solutions were injected into the anterior intraventricular vein using a porcine model and the in vivo gel formation and retention in the vein and tributaries characterized (Fig. 2A). Explanted hydrogels displayed excellent segmental uniformity with minimal differences in equilibrium swelling ratio and gel fraction across the hydrogel length (proximal to distal). It should be noted that the calculated gel fraction does not consider any potential leaching from the gel between the time of injection until explant. The equilibrium swelling ratio provides a measure of the hydrogel network structure and indicated that the gel formation was uniform across the length of the vein (Fig. 2B). Once the hydrogel is injected into the vein and connected to a pacing device, it will remain in that location during the patient's lifetime. We conducted a preliminary safety assessment to confirm that the occlusion of the vein does not result in acute adverse effects on the animal and to evaluate the host response to the gel. Unlike arterial interventions, targeting the venous vasculature offers several advantages: 1) occlusion is well tolerated clinically with no ischemic events[15–17]; 2) lack of vascular remodelling for stable placement; and 3) low venous pressures (<10 mmHg during diastole, <30 mmHg during systole) that are unlikely to dislodge the hydrogel as confirmed in this study[18]. In clinical settings the hydrogel will be delivered to the veins via a catheter; however, for this initial feasibility study, an epicardial incision was made in order to access the vein and deliver the hydrogel. In this specific case, the hydrogel was injected into the anterior

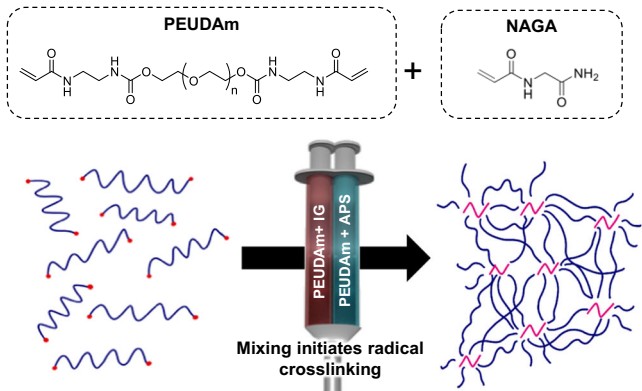

**Fig. 1 | Design of injectable hydrogel electrode.** Redox initiation reaction of polyether urethane diacrylamide (PEUDAm) macromer + N-acryloyl glycinamide (NAGA) delivered using double barrel syringe with a mixing head. Ammonium persulfate (APS) and iron gluconate (IG) are used as initiator and reducing agents.

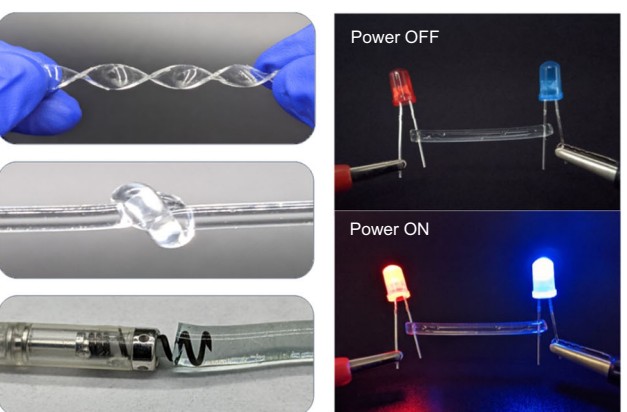

Resulting hydrogels display bidentate hydrogen bonding at netpoints for improved durability. Ionic conductivity is conferred through the inclusion of salt species in the hydrogel precursor solution.

**A** *In vivo* cure of hydrogel in cardiac veins and tributaries

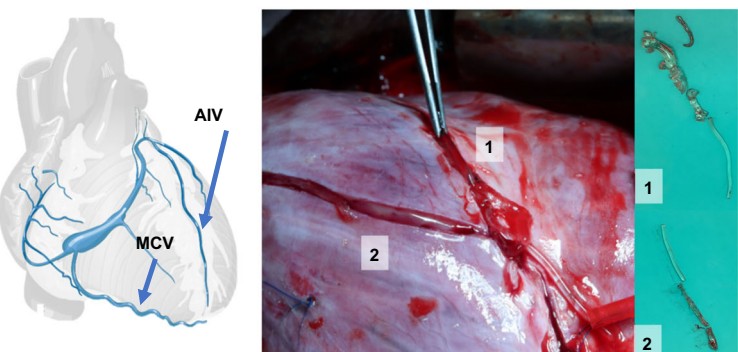

**B** Hydrogel homogeneity assessment postmortem

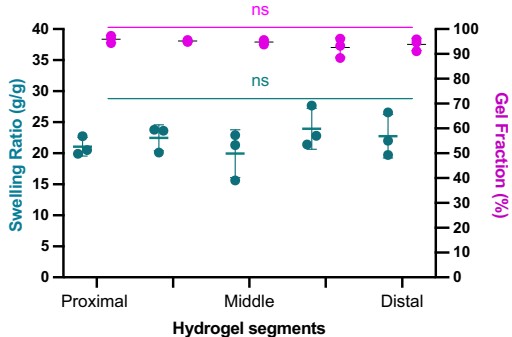

**C** Histological analysis of the AIV and MCV containing hydrogel (H&E)

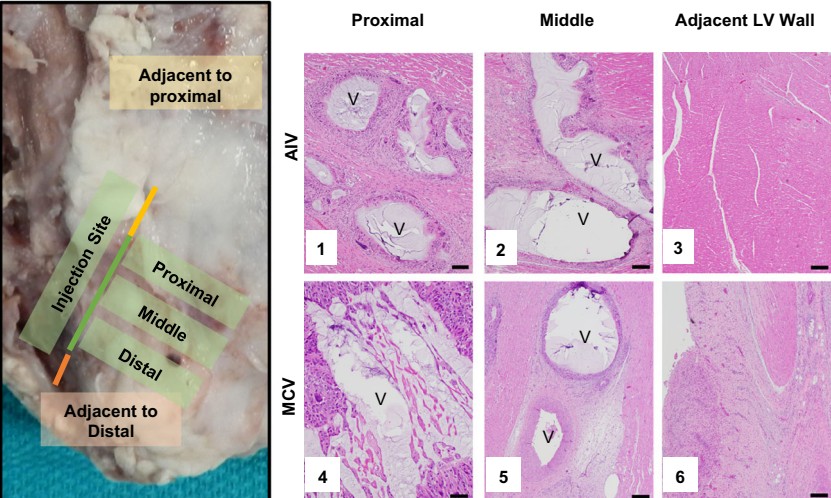

**D** Cardiac functional assessment

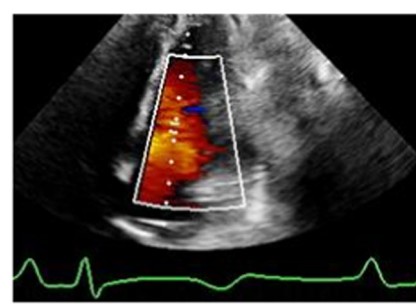

### Ejection Fraction

| Day 0 | 4 Weeks |
| --- | --- |
| 60-70% | 65-69% |

### Cardiac Enzymes: Troponin

| Day 0 | 2 Weeks | 4 Weeks |
| --- | --- | --- |
| 17-121 ng/L | 28-97 ng/L | 45-54 ng/L |

**Fig. 2 | In vivo assessment of injectable electrode in coronary vein of a porcine model. A** Postmortem confirmation of in situ cure of hydrogel in the anterior interventricular vein (AIV) (H1) and tributaries (H2). Coronary vein image from IMAIOS.com. **B** Injected hydrogel homogeneity assessment of gel fraction (pink circles) and swelling ratio (green circles). Means and standard deviation are presented ($n = 3$ independent animals, 5 hydrogel segments per animal). ns = nonstatistical difference between the segments ($p > 0.05$) determined with an ordinary two-way ANOVA. For multiple comparisons, a Tukey's test with a single pooled variance was used. Source data are provided as a Source Data file. Exact *p*-values were provided in the source data file. **C** Host response after 4 weeks (P-2165, Animal AIV 2) implantation using slices of the location proximal to the injection site, middle of the vein, and the control from an alternate section of the anterior left ventricle (LV) wall. The response includes damage induced at the hydrogel injection site, with 1) Arrows are presented in the images on the Supplementary information

with foreign body giant cell reaction). Mid branch of the hydrogel injection show 2) focal replacement fibrosis, and fibrosing epicarditis. The control images at an alternate section of the anterior wall indicated 3) preserved myocardium. Histology from the middle cardiac vein (MCV) is also shown after 2 weeks of implantation, with 4) fibrosing epicarditis) with foreign body giant cell reaction. Mid branch of the hydrogel injection show 5) mild focal replacement fibrosis and fibrosing epicarditis. The control images at the distal branch indicated 6) preserved myocardium with only fibrosing epicarditis with focal extension into myocardium. Scale bars are 200 μm for 1,2,3,5, and 6. Scale bar is 50 μm for 4. Three biological replicates were performed on the AIV and one for the MCV to confirm the results (Supplementary Figs. 5–8). V = vein. **D** High sensitivity troponin measurements from blood draw and measured ejection fraction from cardiac echocardiography. The echo image from day of termination for Animal AIV 2. For full videos, see Supplementary Movie 1–3.

intraventricular vein (AIV) and retained for four weeks ($n = 3$) and into the middle cardiac vein (MCV) and retained for 2 weeks ($n = 1$). Tissue slices of location proximal, adjacent, and distal to the injection site, were excised and analyzed to assess the extent of cardiac injury and inflammation after two (Supplementary Fig. 5) and four weeks, respectively (Fig. 2C, Supplementary Figs. 6–9). There was no evidence of myocardial necrosis on histopathology and no evidence of damage to the left ventricular myocardium. Mild perivascular and interstitial fibrosis was observed across all locations, with chronic epicarditis closer to the incision/injection sites (Fig. 2C). Chronic inflammation with foreign body giant cell reaction was observed in the hydrogel-containing vessels. This reaction extended into the myocardium immediately surrounding the vessel. Replacement fibrosis was observed in some of these areas immediately surrounding the vessel. There was no diminution in myocardial contractility and no evidence

of regional wall motion abnormalities at four weeks compared to baseline imaging (serial echocardiograms). The overall QRS morphology was also preserved 4 weeks after injection into the AIV (Supplementary Figs. 10–12) and similar two weeks after injection into the MCV (Supplementary Fig. 13). In addition to cardiac function, cardiac enzyme levels were monitored on the day of injection and the weeks after injection (Supplementary Table 1). Troponin levels before the procedure ranged from 17–121 ng/L with no significant change at the end of the studies (MCV, 2 weeks: 28–97 ng/L; AIV, 4 weeks: 45–54 ng/L), indicating that the hydrogel did not induce ischemia or myocardial necrosis (Fig. 2D). No evidence of platelet accumulation and clot formation was observed in the vessels after hydrogel formation. No thrombotic activity was expected given that there was no blood flow after the epicardial vessel was occluded with the injected hydrogel. Upon necropsy after 4 weeks, the lungs were clear of clots or other

signs of adverse events which supports this observation, as any venous embolization of materials would manifest in the lung.

## Hydrogel electrode mimics physiologic conduction by capturing midmyocardial tissue

The fundamental assumption in development of the injectable hydrogel electrode is that it should be able to pace the heart from the cardiac veins. The clinical workflow for the proposed hydrogel electrode includes: hydrogel precursors solutions delivered using a dual lumen catheter to fill coronary veins and tributaries that span the myocardium near the scarred tissue with redox-initiated crosslinking providing rapid cure of the ionic hydrogel (Fig. 3A); subsequent connection to a pacemaker lead to the ionic hydrogel electrode provides increased tissue contact across the myocardium (Fig. 3B); wavefront activation of the myocardium along the hydrogel electrode reduces the energy required for defibrillation (Fig. 3C). We tested the injectable hydrogel electrode in a porcine model to simulate clinical deployment.

At a pulse width of 5 ms, unipolar pacing thresholds were measured to be $2.2 \pm 1.2$ mA for the metal electrode, $2.3 \pm 0.6$ mA, $1.5 \pm 1.28$ mA for an epicardial hydrogel point source (0.8 cm disk), $1.5 \pm 1.3$ mA for the epicardial hydrogel line source and $2.7 \pm 1.8$ mA with the hydrogel in the AIV. There was no significant difference in the capture thresholds of the different electrode configurations at 5 ms and 10 ms pulse widths. Pacing thresholds measured at other pulse widths are detailed in (Supplementary Fig. 14 and Supplementary Table 2). Two-way ANOVA revealed significant differences only for capture thresholds with pulse widths below or equal to 1 ms. Epicardial pacing using a metal electrode, hydrogel point, or hydrogel line, created an inverted QRS morphology indicative of a deviation from normal conduction, Fig. 3D. Remarkably, the QRS morphology by pacing through the hydrogel in the AIV was comparable to the QRS at sinus rhythm. In particular, the upright deflections in leads I, II, and aVL; biphasic deflections in leads III and aVF; and inverted deflections in aVR are remarkably similar between sinus rhythm and hydrogel-mediated

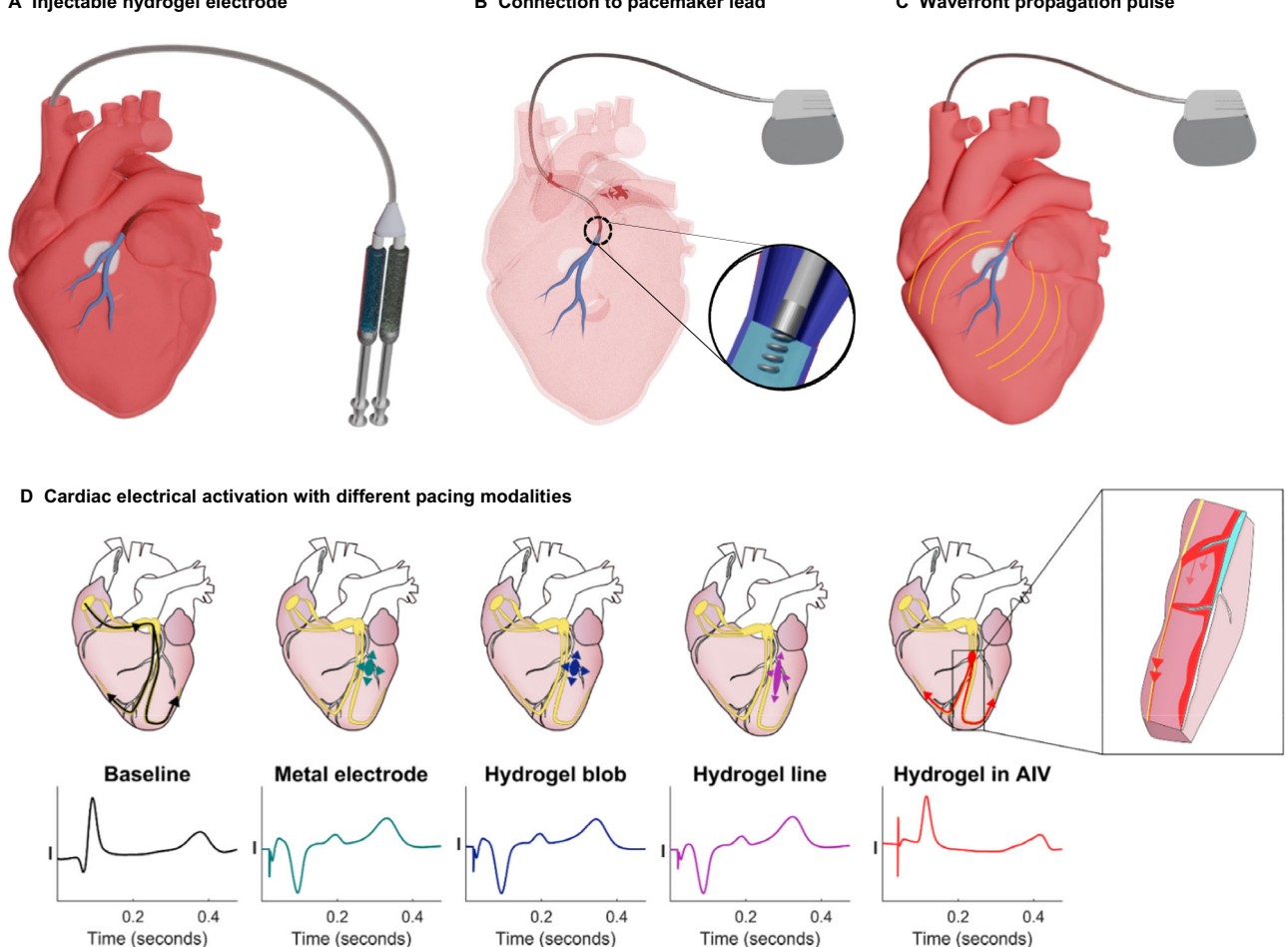

**Fig. 3 | Schematic of injectable hydrogel system to transform coronary veins into flexible electrodes that capture inaccessible cardiac tissue. A** Hydrogel precursors solutions are delivered using a dual lumen catheter to fill coronary veins and tributaries that span the myocardium near the scarred tissue. Upon mixing of the two hydrogel solutions in the coronary veins, redox-initiated crosslinking results in rapid cure of the ionic hydrogel. **B** Subsequent connection to a pacemaker lead to the ionic hydrogel electrode provides increased tissue contact across the myocardium. **C** Wavefront activation of the myocardium along the hydrogel electrode reduces the energy required for defibrillation. **D** Lead I of surface ECG showing electrical activation at baseline (sinus rhythm) and when paced. Peak of the baseline QRS morphology was matched with the peak of QRS morphology obtained from pacing with hydrogel in AIV (Top, left to right) Baseline: Activation

during sinus rhythm starts from the sinoatrial node and travels through the atrio-ventricular node into the Purkinje fibers in the ventricles. Pacing the myocardium directly causes the QRS morphology to invert indicative of deviation from normal conduction. Point pacing with metal electrode and hydrogel point creates a dog-bone shaped activation that radiates outward from the source of the electrical stimulus. Hydrogel line pacing also created an inverted QRS morphology which indicates direct myocardial activation. Pacing from hydrogel in the AIV displayed clear capture with a QRS morphology existing after every pacing spike. No observed inversion of the QRS morphology and a short isoelectric window before initial activation indicates possible capture of the deep septal bundle branches and Purkinje fibers (inset). Source data are provided as a Source Data file.

pacing. Point pacing with metal electrodes did not yield such morphological similarities. It should be noted that the QT intervals were not identical to baseline in all cases of AIV hydrogel pacing. The primary motivation in analyzing the surface morphologies was to assess mid-myocardial capture, using native conduction system capture as a surrogate. The extensive literature in the field has focused almost exclusively of QRS morphology analysis[19–22]. T-wave analysis is not used clinically for this purpose. Surface 12-lead traces and corresponding analysis for each condition is included in the Supplemental Information (Supplementary Figs. 15–29). It can be appreciated that QRS morphologies and T-wave morphologies are far more similar with AIV hydrogel pacing as compared to all other conditions. No inversion of the QRS morphology and the short isoelectric window before initial activation indicate the hydrogel electrode captures the deep septal bundle branches and Purkinje fibers for the first time[23,24].

### Pacing via hydrogel electrode normalizes tissue activation across heterogeneous myocardium

Electroanatomical mapping of impulse propagation in a porcine ablation model was used to study the effect of pacing from a point source and the hydrogel electrodes on conduction across heterogenous tissue. Briefly, ablation was performed on the epicardium of a pig heart near the AIV to disrupt the native conduction and mimic scarred myocardium conditions. Post-mortem dissection was used to demonstrate the depth of the scar formation following ablation, Fig. 4A. Point pacing demonstrated a delayed and heterogeneous (focal) activation wavefront after ablation which was attributed to ablation lesion formation, Fig. 4B. The hydrogel was injected and cured in the AIV as described above and subsequent voltage mapping confirmed that the hydrogel electrode increased tissue activation area, Fig. 4C. Moreover, the electroanatomical mapping demonstrates for the first time that the activation wavefront from the hydrogel reaches the midmyocardium and endocardium much earlier in the ablation model than point pacing. The second critical observation is the broad swatch of early activation observed across the midmyocardial and endocardial aspects with AIV hydrogel pacing. Supplementary Supplementary Figs. 30–32 confirm these findings in two independent replicates.

### Discussion

Re-entry is the leading cause of lethal arrhythmias[1]. Its most common mechanism is delayed cardiac conduction in scarred tissue,

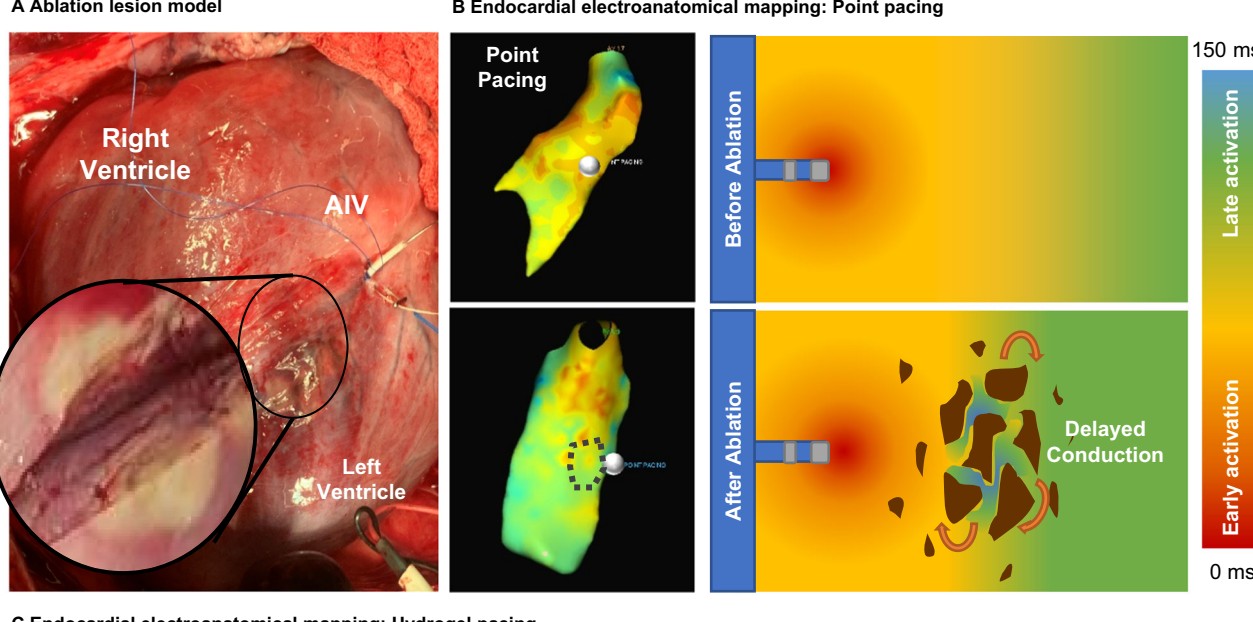

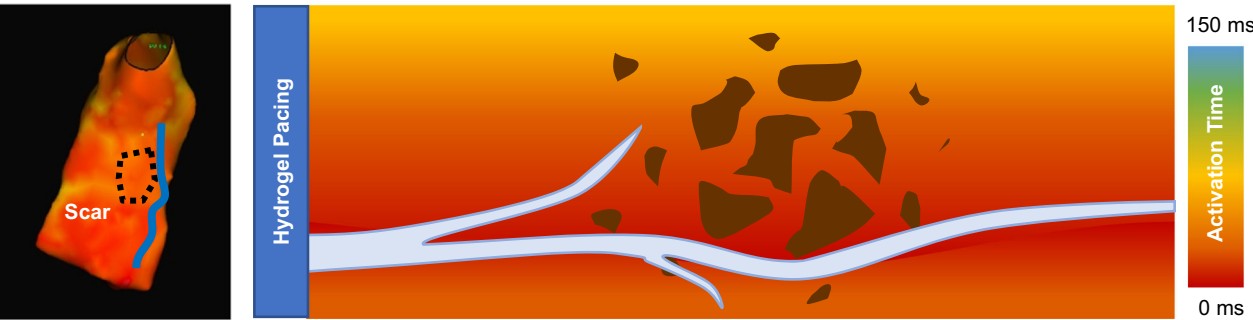

**Fig. 4 | Cardiac electroanatomical mapping studies in a porcine ablation model. A** Image of cardiac ablation near the AIV with substantial depth of the ablation lesion formation. **B** Endocardial electroanatomical mapping of point pacing before and after ablation with red indicating early capture and blue indicating late capture. Point pacing location is indicated by a small white circle and region of lesion by dotted black line. Illustration demonstrating how point pacing allows for development of re-entrant circuits (indicated by the orange arrows) due to heterogenous capture and activation around the lesion (brown area).

**C** Endocardial electroanatomical mapping of hydrogel pacing after ablation with red indicating early capture and blue indicating late capture. Hydrogel electrode location is indicated by a blue line and region of lesion by dotted black line. Illustration demonstrating how the hydrogel in the vein generates a planar wavefront through long, linear areas of capture that prevent re-entry by normalizing conduction across heterogenous tissue. Scale represents the tissue activation time from 0 to 150 ms after pacing.

predominantly observed after coronary artery occlusion during a myocardial infarction[25]. The ability to directly pace and capture these regions would enable targeted correction of the local activation delay; however, development and testing of such an approach has been limited by the inability to directly access and pace these regions. Given that the venous structures drain the same region where the arterial occlusion caused ischemic damage, these vessels and their tributaries provide a direct approach to the scarred tissue. However, there are no pacing leads small enough to navigate these smaller tributaries and epicardial approaches have limited efficacy due to the greater mass and thickness of ventricular tissue. In this work, we developed an injectable hydrogel electrode that can fill coronary veins to convert them into flexible electrodes that directly pace the inaccessible mid-myocardium. Hydrogels are being explored as potential flexible electrodes due to their tunable properties and ease of incorporating conductive elements[26]. An important criterion for translation was identifying a system that was biostable, biocompatible, and can be used with current endovascular approaches. We designed a hydrogel chemistry that could meet the multifaceted needs of this application. The PEUDAm hydrogels displayed stiffness similar to the myocardium, necessary to avoid adverse tissue responses[27], and increased biostability as compared to traditional PEG diacrylate. These hydrogel electrodes also needed adequate conductivity to ensure stimulation of the myocardium along the length of the hydrogel. Many of the conductive polymers are not suitable as an injectable electrode due to insolubility, toxicity, or processing limitations. Our ionic hydrogel utilizes salts added to the precursor solutions to confer conductivity without detriment to other hydrogel properties, solution viscosity, or cure rate. The resulting ionic hydrogel electrode achieved conductivity more than double target myocardium values. The higher conductivity of the hydrogel with respect to tissue will allow ions to move across the gel much faster than the tissue, creating a conduction highway similar to Purkinje fibers as well as multipoint activation along the length of the hydrogel. We refer to this as a planar activation and utilized the ablation model to test that it can be used to potentially circumvent the conduction delays caused by the tissue heterogeneity across scar tissue and terminate re-entrant currents around the scar at low energies. This is not possible using a single-point activation with a metal electrode.

Regarding clinical implementation and safety, our approach is designed to integrate with current clinical workflow using a standard pacing system as shown in Fig. 3. Our two-component system relies on delivering hydrogel precursor solutions through a dual lumen catheter to provide direct access to the diseased regions with relative precision and ease. Mixing of the two solutions upon injection into the vein activates redox-initiated crosslinking of the gel for rapid in situ cure. In contrast to other systems that rely on reversible bonds or physical crosslinking, our approach relies on inducing chemical crosslinking between the polymer chains via in situ radical polymerization to ensure long-term stability of the hydrogel. This system is highly tunable and components can be adjusted to achieve a broad range of hydrogel properties and cure rates without changing the chemistry of the macromers. Assessment of hydrogel uniformity confirmed successful in vivo deployment with acute cardiac pacing. A separate study confirmed the retention of conductivity after implantation; however, chronic pacing studies require additional large animal studies with an implantable pacemaker.

Clinical experience together with the current data supports the potential of transvenous catheter-based delivery of the hydrogel electrode to the AIV through the subclavian vein or internal jugular. Although there are currently no commercially available catheters that are suitable to deliver the in situ curing injectable hydrogel electrode, there are dual lumen catheters that can be modified for this purpose. A dual lumen catheter is currently under development that can 1) provide controlled delivery of two precursor solutions at equal flow and

volume, 2) ensure homogeneous mixing, 3) prevent the solution from clearing from the vein due to venous return. Our current prototype is based on a dual lumen design that separately delivers the precursor solutions with a mixing head at the distal tip of the catheter to ensure homogenous mixing, Supplementary Fig. 33. A dual injection system was developed for the proximal hub of the catheter to ensure the solutions are injected at the same flow rate and an occlusion balloon proximal to the distal tip, with an accompanying lumen to allow for air- or saline-mediated inflation and deflation, was included to prevent the precursor solutions from clearing prior to cure due to venous flow. This prototype catheter is under development concurrent to the hydrogel development. As such, we do not expect that it will limit the clinical translation of the proposed hydrogel electrode.

By delivering the hydrogel to the veins, we avoid arterial occlusive ischemia. Furthermore, there may be some concern regarding cardiac engorgement consequent to the occlusion of the epicardial veins with the hydrogel electrode. However, extensive clinical experience strongly argues in favor of the safety of this practice—hundreds of thousands of occlusive leads are placed in these same vessels each time a CRT device is implanted—with no reports of complications. Longstanding literature also supports the safety of total occlusion of the coronary sinus (into which all venous tributaries drain)—a vessel much larger than the branch vessels we will be targeting[15–17]. There is a theoretical risk of edema due to reduced venous return after occlusion; however, we expect minimal effect due to established venous remodeling under similar conditions. The pilot large animal data shows that occluding the epicardial venous tributaries was associated with minimal, if any, myocardial necrosis or loss of cardiac function. Histopathologic studies demonstrated myocardial inflammation that was increased at the regions directly instrumented in this open chest model. Clinically, such a traumatic delivery system is unlikely to be utilized. Our delivery system is currently being modified for transcatheter delivery using a dual lumen catheter. Gaining access to these venous branches using a closed chest transcatheter approach is a standard skill set utilized by clinical electrophysiologists when placing a cardiac resynchronization device (a pacemaker with a lead in the ventricular branches of the epicardial venous system). Such an approach will reduce the observed inflammatory response at the injection site. The extent of inflammation in the myocardium and the replacement fibrosis observed immediately surrounding the vessels containing the hydrogel in the chronic porcine studies warrants further investigation to determine its origin and temporal stability. It cannot be overemphasized that in no case was there any evidence of myocardial necrosis. A separate study examined the cytocompatibility of the redox hydrogel as well as the host response in a 4-week subcutaneous implant study. Although these studies support the biocompatibility of the hydrogel, additional long-term studies are needed to fully characterize the chronic host response to the hydrogel electrode and potential vessel remodeling. In addition, future studies will explore vessel dilation and mechanical disruption of the vessel with hydrogel fill ratios and the corollary effect on remodeling. Most notably for early demonstration of the safety of this approach, cardiac function was preserved and cardiac enzymes at the end of the study were within normal ranges, indicating a lack of myocardial necrosis or serious disease state. Furthermore, our preliminary data indicated no gross emboli in the lungs which indicates that full occlusion with the hydrogel does not carry the risk of platelet accumulation or embolism. Future studies will characterize the expected collateral venous remodeling and cardiac function over longer periods to fully assess the safety of this approach.

With this injectable hydrogel electrode, we are able to explore the possibility of *prevention* of re-entry by directly capturing zones of delayed conduction, thus eliminating one of the necessary conditions for re-entry to arise. Unlike past efforts that used conductive hydrogels or other materials as scaffolds to regenerate myocardial tissue[28–30],

these conductive hydrogels will interface with current cardiac resynchronization devices and act as biostable, flexible electrodes to provide an immediate intervention to prevent VA. By injecting into the corresponding venous branches with the conductive hydrogels, electrophysiologists can target specific locations corresponding to each patient's scar location(s), enabling patient-specific cardiac resynchronization paradigms involving increased capture areas compared to currently available lead-based systems by virtue of the mid-myocardial nature of the hydrogel electrode. The advantage of direct mid-myocardial tissue capture is not limited to prevention of re-entry. Recent progress in atrial defibrillation technology has demonstrated that using multiple electrodes to deliver multi-stage electrotherapy can reduce the energy required defibrillation[31]. Delivering multiple low-energy pulses over a larger area has shown benefits for terminating fibrillation with energies below pain thresholds[32–34]. Moreno et al. performed an ex vivo study with epicardial line electrodes that generated a planar activation wavefront that depolarized a large ventricular area at energy levels below the human pain threshold[35]. These reports indicate that planar wavefront propagation from the hydrogel electrode as it spans the mid-myocardium also provides the opportunity for painless ventricular defibrillation. Other multi-site pacing approaches to prevent and treat VA rely on epicardial placement of flexible electrodes and are not used for the treatment of cardiac defibrillation. Rather such pacing therapy is used for treatment of monomorphic ventricular tachycardia (but no fibrillation). The lack of a minimally invasive deployment strategy of these hydrogels as flexible electrodes represents a large translational challenge for this technology. Furthermore, epicardial stimuli are less effective for defibrillation due to the greater mass and thickness of ventricular tissue. In contrast, this endovascular approach provides unprecedented simultaneous multi-site pacing in the mid-myocardium while also integrating with current clinical practices. In our preliminary in vivo studies, once the hydrogel was injected into the AIV, we observed uniform capture along the vessel both pre and post ablation as well as increased capture area as compared to point pacing. This data supports the idea that the planar wavefront will persist without causing re-entrant activity.

Cardiac pacing studies using these injectable hydrogel electrodes in a large animal model provided proof-of-concept for this approach. Although we did not assess the effect of such pacing in diseased myocardium, our work demonstrates for the ability to directly pace these erstwhile unapproachable heart regions using current commercial technology. The evidence to support this is the nearly identical nature of surface electrocardiographic morphologies of the paced and native heartbeats. A similar effect normalization of QRS vectors has been observed in clinical reports of deep septal pacing[36–38]. Furthermore, there is a marked latency between the pacing stimulus and initiation of the surface ECG. This is because the pacing output at least in part captures and conducts via the native conduction system before exiting into myocardial tissue. Thus, there is a delay between the pacing stimulus and onset of the ECG signal. These features were reproducibly observed and are strongly suggestive of direct mid-myocardial tissue capture, something that has been technologically vexing and, to the best of our knowledge, unreported. Following this exciting finding, we tested our central hypothesis that simultaneous stimulation of wide areas of ventricular tissue can diminish the influence of tissue heterogeneities (e.g. scarring) that cause delayed cardiac electrical conduction and re-entrant fronts. A porcine myocardial infarct model with increased incidence of ventricular arrhythmias is currently under development for full evaluation of the clinical benefit of the proposed technology. To provide proof-of-concept assessment of the effect of tissue heterogeneity on conduction, we compared point and hydrogel electrode pacing in the commonly-accepted ablation model. Electroanatomical mapping clearly demonstrated the delayed conduction after ablation with point pacing and that the hydrogel electrode dramatically increased the activation area to resolve this conduction delay. Although this model does not fully replicate the effect of scarring after myocardial infarct, it does demonstrate that the hydrogel electrode shows much more rapid conduction and minimized the effect of tissue heterogeneity on conduction velocity in at least limited regions of myocardial injury. Given the well-established fundamental link between tissue heterogeneity and ventricular arrhythmias, the hydrogel electrode has demonstrated the potential to terminate re-entry well below the pain threshold. A method for painless defibrillation and potential prevention of arrhythmias would revolutionize cardiac rhythm management.

## Methods

All animal procedures complied with regulations and were approved by the IACUC at the Texas Heart Institute. The following protocols were approved to conduct the animal studies detailed in this section: 1) Protocol# 2021-08: Conductive Hydrogel for Myocardial Pacing based Therapy in a Chronic Porcine, and 2) Protocol# 2022-01: Pacing-Mediated Restoration of Conduction Across Scarred Myocardium for Prevention of Cardiac Arrhythmias via Intravascular Injection of Conductive Hydrogel. Details about the animal specimens are provided in Supplementary Table 3. Although both female and male animals were used in the study, this pilot data was not suitably powered to consider sex as a biological variable.

### Materials

All materials were purchased from Sigma Aldrich unless otherwise indicated.

### Synthesis of hydrogel precursors

A polyether urethane diacrylamide (PEUDAm) macromer was synthesized in a three-step process with a PEG-diamine intermediate, Supplementary Fig. 1. Briefly, a PEG diol (Sigma, No. 81275, Mn = 17600, 1 equiv.) was first functionalized with carbodiimidizole (CDI, Alfa Aeser No. A14688, 15 equiv.) and purified prior to subsequent reaction. Ethylenediamine (15 equiv.) was then added to the PEG-CDI solution (12 wt%, 1 equiv.) in anhydrous dichloromethane (DCM from Sigma No. 34856) and the product was precipitated in ice-cold ether (Sigma, No. 673811) and collected via suction filtration. This product was then functionalized with acrylamide endgroups by reacting with acryloyl chloride (Sigma, No. 549797, 4 equiv.). The product was precipitated in 10x volume ice-cold diethyl ether and dried at atmospheric pressure overnight, then briefly under vacuum. Functionalization of PEG macromer at each step was determined by analyzing proton nuclear magnetic resonance ($H^1$ NMR) spectra collected on a 300 MHz Bruker NMR, Supplementary Figs. 2–S4. $H^1$ NMR (CDCl3): PEG-CDI: $\delta$ = 8.12 (s, 2H, -N-CH-N-), 7.40 (s, 2H, -N-CH = CH), 7.02 (s, 2H, -CH = CH-N), 4.51 (m, 4H, -O-CH2-CH2-), 3.70 (m, 1720H, -O-CH2-CH2-). PEG-EDA: $\delta$ = 4.18 (t, 4H, -O-CH2-CH2-), 3.60 (m, 1720H, -O-CH2-CH2-), 3.19 (m, 4H, -CH2-CH2-NH), 2.78 (m, 4H, -NH2-CH2-CH2-), 3.70 (m, 1720H, -O-CH2-CH2-). PEUDAm: $\delta$ = 6.94 (broad s, 2H, -C-NH-CH2-), 6.28 (dd, 2H, H2C = CH-C-), 6.17 (m, 2H, H2C = CH-C-), 5.85 (broad s, 2H, -C-NH-CH2-), 5.61 (dd, 2H, H2C = CH-C-), 4.20 (m, 4H, -H2C-CH2-O-), 3.65 (m, 1720H, -O-CH2-CH2-), 3.34 (m, 4H, -CH2-CH2-NH-). *N*-acryloyl glycinamide was synthesized according to established protocols with minor variations[10].

### Study preparation of animal for in vivo testing

Yorkshire pigs between 40 kg and 50 kg were used for these studies. Full details regarding animal sex and age are provided in supplementary information. In preparation for the study, the animal was fasted for 12–24 hour prior to surgery; water was provided ad libitum. During surgical preparation, an intramuscular injection of Telazol (4-6 mg/kg) and Atropine Sulfate (0.02–0.05 mg/kg) was applied for sedation. Isoflurane (0.5–3.0%) was then administered via facemask to induce general anesthesia. The animal was tabled in a supine position and intubated. An intravenous catheter was established for drug delivery

and the animal was transported to the operating room. Appropriate preoperative analgesics such as Buprenorphine HCL (0.005–0.1 mg/kg) and/or Flunixin Meglumine (1.1–2.2 mg/kg) were administered intravenously. Anesthetic agents were given to effect such that adequate levels of anesthesia are achieved without exposing the animal to unnecessary pharmacological effects. The animal's vitals were constantly monitored using a 12-lead ECG for the entirety of the procedure.

## Hydrogel fabrication and in vivo deployment

Hydrogels were fabricated with the PEUDAm macromer (20 wt%), NAGA monomer (1 wt%, BDL Pharma, custom order) and a combination of redox initiators ammonium persulfate (APS, 0.75 mM, Sigma, No. 248614) and iron gluconate (IG, 1.5 mM, Sigma, No. 44948) in saline solution (1.35% NaCl). Hydrogel solutions were loaded into disposable double-barrel syringes with half the solution containing APS and the other half containing IG. To determine conductivity from the impedance analysis, the hydrogel constructs ($n = 8$) were placed between two electrodes with an alternate current sweep from 1 to $10^6$ Hz. Samples were tested on an impedance analyzer equipped with an enhanced controlled environment sample holder from BioLogic (ID CESH-e, Electrodes 097-150/TP20). Excess water (or blood from implanted hydrogels) was removed from the swollen samples using a kimwipe before testing. The resistance (R) of the sample was obtained when the impedance plateau in the bode plot. Conductivity (σ) was calculated as the inverse of the resistance multiplied by the gel thickness (L) and divided by the area (A), σ=L/(R$x$A).

For initial hydrogel electrode deployment, pigs ($n = 3$) were sedated and prepared as described above and a median sternotomy was performed to expose the anterior surface of the heart. The anterior intraventricular vein (AIV) branch that would eventually be utilized for injection was identified prior to pacing, and four decapolar catheters were sutured around the target site in a square for sensing the local epicardial electrograms. The pacing target was selected close to the expected injection site on the AIV. Unipolar cathodal pacing using a bare metal electrode, hydrogel point, and hydrogel line were performed with the indifferent electrode placed on the animal's chest. Three different modes of pacing were attempted—point pacing, line pacing, and pacing with the hydrogel in the AIV. Hydrogel precursors were prepared as described above and loaded into a double-barrel syringe with a mixing head. To prepare the hydrogel for point pacing, approximately 500 µL of hydrogel was dispensed into a 2 mL tube containing an exposed stainless-steel wire such that that the gel cured (~8 mm diameter) around the end of the wire. To prepare the hydrogel line source, the hydrogel precursors were injected into a 14 G angiocatheter (BD, Angiocath No. 382268) and allowed to cure for 2 minutes to generate a cylindrical hydrogel line source for epicardial pacing (1.6 mm in diameter, 8 cm in length). The minimum current required to achieve reliable pacing capture (called capture threshold) at a constant pulse width (0.5, 1, 5, and 10 ms) was recorded for each pacing mode, yielding different strength-duration curves. After completion of epicardial pacing, the hydrogel was injected into the AIV, using a 14 G angiocatheter inserted directly into the AIV with a suture at the proximal end to minimize bleeding and backflow of the hydrogel precursor solutions upon injection. A double barrel syringe with a mixing head was loaded with 2 mL of hydrogel precursor solutions and attached to the angiocatheter via the luer lock system described above. After allowing the gel to cure for ~2 minutes, the catheter was removed and a pacing lead was attached to the hydrogel at the proximal end using an alligator clip. Capture threshold and strength-duration curves were again assessed with the intravascular hydrogel. At the end of the study, all animals were humanely euthanized per standard IACUC approved practice. After euthanasia, the AIV was carefully opened using a scalpel and the cured hydrogel was carefully excised from the AIV for inspection. The hydrogel was then sectioned into five segments from the proximal location to the injection site to the distal location in the ventricle and characterized for sol fraction and swelling ratio. To determine the sol fraction, the segments were submerged in deuterium oxide overnight. The supernatant was analyzed using NMR to quantify the amount of PEG versus a standard amount of isopropanol ($W_{sol}$). To measure the equilibrium swelling ratio, the samples were subsequently swollen in DI water for 24 hours and weighed ($W_s$). The specimens were then dried under a vacuum for 24 hours and weighed ($W_D$). The equilibrium mass swelling ratio was calculated as Q=$W_s$/$W_D$. The gel fraction was calculated as $W_D$ /($W_D + W_{sol}$).

## Safety assessment in a porcine model

A porcine model was selected for safety and feasibility testing due to the striking similarity between the porcine and human coronary artery and venous systems. The same main veins and arteries seen in human hearts follow the same path and structure in pig hearts. Most importantly, the intraventricular septum circulation (studied in the work presented) are very similar. There are some slight physiological differences, such as the left anterior descending artery lying slightly rightward in pigs compared to humans, and the left coronary ostia having a slightly different angle of entry compared to humans[39]. Bsed on these similarities, Yorkshire pigs were utilized for the chronic assessment of the safety of the injected hydrogel vein occlusion of the AIV ($n = 3$) and the (MCV, $n = 1$). The pigs were sedated and prepared as previously described with vitals and ECG recorded and monitored throughout the study. For the AIV, a median sternotomy was performed on the pig to expose the anterior surface of the heart. The sterile hydrogel precursor solutions, prepared as described above, were injected via the double-barrel syringe into the vein. The gel was allowed to cure for 5 minutes before catheter removal. For the MCV, a thoracotomy was performed to expose the left side of the heart. The largest vein in that window, the MCV, was identified by the surgeon and accessed using a 16 G angiocatheter (BD, Angiocath No. 381157). The sterile hydrogel precursor solutions, prepared as described above, were injected via the double-barrel syringe into the vein. The gel was allowed to cure for 5 minutes before catheter removal. A single prolene suture was placed at the injection site to minimize any potential bleeding and as a marker for histological analysis after 2 (MCV, $n = 1$) and 4 (AIV, $n = 3$) weeks. Echo cardiograms and blood enzymes were drawn at the beginning and end of the study to assess myocardial changes. The animal was allowed to recover and was observed for 2 and 4 weeks prior to euthanasia. The animals were prepared for termination, and ECGs, echocardiogram, and enzyme blood draws were repeated before termination. After euthanized was complete, the heart was removed and placed in fixative for histological analysis. A small section of hydrogel was also removed for sol fraction and swelling ratio analysis prior to fixation. The heart was sectioned along the length of the hydrogel injection to include the proximal vein for site of injection, the mid-vein, and the distal vein, all approximately 2 cm apart. Paraffin-embedded sections were stained with either hematoxylin and eosin, Masson's Trichrome, or Masson's Pentachrome to assess cardiac injury and inflammatory responses of the tissue to the chronic hydrogel implant.

## Cardiac electroanatomical mapping in a porcine ablation model

Yorkshire pigs were sedated and prepared ($n = 3$) as previously described with vitals and ECG recorded and monitored throughout the study. A midline sternotomy was performed to expose the heart. A baseline sinus rhythm activation map was measured on the epicardium, endocardial left side, and endocardial right side utilizing the EnSite Precision Cardiac Mapping System (Abbot St. Jude, IL, EnSite Precision Module Software H702496 EnSite Precision Module Hardware H702470). After sinus mapping, the heart was paced at the expected site of hydrogel injection using a temporary pacing lead and mapped again at the same locations described above. Pacing was

performed using the BARD Electrophysiology system (Boston Scientific, MA, M004 2001205 0 Isolation Transformer). The hydrogel electrode was then injected and paced as described previously. Activation was mapped during hydrogel pacing in the same locations as above. After mapping these three conditions, ablation was performed distal to the injection site near the AIV to mimic the conduction delay caused by cardiac scarring. Approximately 5 ablations were performed in a lateral line at 50 W for 30 s with saline irrigation using the 8Fr THERMOCOOL Catheter (Biosense Webster, CA, No. NI75TCJH) with CoolFlow irrigation pump (Biosense Webster, CA, Model: M-5491-00) and Stockert 70 RF Generator System (Biosense Webster, CA, discontinued). After ablation, the three pacing conditions (sinus rhythm; temporary pacing lead; hydrogel) were repeated along with their associated mapping procedures (n = 3). The animals were terminated upon completion of the procedure.

### Statistical analysis

Quantitative characterization of hydrogel properties is expressed as mean ± standard deviation. Statistical comparisons were made using analysis of variance (ANOVA) for multiple comparisons with the Tukey test. The Tukey test compared all possible pairs of means based on studentized range distribution (q). The normality of the data was assessed using the Kolmogorov–Smirnov test and the Shapiro–Wilk test. Computations were performed using GraphPad Prism version 9.0.2 at the significance levels of $p < 0.05$.

### Reporting summary

Further information on research design is available in the Nature Portfolio Reporting Summary linked to this article.

## Data availability

All data supporting the findings of this study are available within the article and its supplementary files. Any additional requests for information can be directed to, and will be fulfilled by, the corresponding author(s). Source data are provided with this paper.

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

## Acknowledgements

The authors are grateful for the contributions of the THI Animal Lab staff, particularly Dr. Abdelmotagaly Elgalad and Dr. Angel Moctezuma-Ramirez, as well as the THI Pathology Lab, particularly Dr. Pamela Potts and Dr. Ana Segura. We also thank Dr. Lynd (The University of Texas at Austin) for the use of the EIS and the Center for Electrochemistry and Texas Materials Institute for the use of the four-point probe and rheometer, respectively. The authors also thank Christina Waldron for the preparation of Fig. 3A–C. Funding was provided by the National Institutes of Health, grant number R01 HL162741 (E.C.H., M.R.), Ford Pre-Doctoral Fellowship, administered by the National Academy of Science, Engineering and Medicine (G.J.R.R.), Ford Dissertation Fellowship, administered by the National Academy of Science, Engineering and Medicine (G.J.R.R.), Office of Vice President for Research, The University of Texas at Austin (E.C.H.), The Roderick D. MacDonald Research Fund Award 19RDM004 (MR), The Sultan Qaboos Chair in Cardiology at the St. Luke's Foundation (M.R.).

## Author contributions

Conceptualization: E.C.H., M.R. Methodology: G.J.R.R., M.J., A.P., S.B., D.B. Investigation: G.J.R.R., M.J., A.P., S.B., Visualization: G.J.R.R., M.J., E.C.H. Supervision: E.C.H., M.R. Writing—original draft: G.J.R.R., A.P., M.J., D.B., E.C.H., M.R. Writing—review and editing: E.C.H., M.R., A.P.

## Competing interests

E.C.H., A.P., M.R., M.J. have filed a patent application entitled: "Electrically conductive hydrogels usable as pacemaker lead extensions, apparatus for delivery of a hydrogel into the venous system, and methods of treating ventricular arrhythmia with electrically conductive hydrogels injected in the venous system", WO2021046441A1. E.C.H., M.R., A.P. are founders of Rhythio Medical which seeks to commercialize the hydrogel electrodes. All remaining authors declare no conflict of interest.
