## [Peer Review File · Nature Communications]

REVIEWER COMMENTS

Reviewer #1 (Remarks to the Author):

The novelty of the approach with the initial experiments are interesting. The authors responded to earlier comments recognizing the need for additional experiments and development as well as longer-term follow-up.

Reviewer #2 (Remarks to the Author):

Given the journal transfer, I do not think additional data is required. However, text changes are needed per previous comments to more accurately reflect the data presented in the manuscript:

The added limitation at the end of the manuscript is good, but all mention of testing the hydrogel in scarred myocardium or labelling ablated tissue as “scar” needs to be removed. The tissue was acutely evaluated after ablation. It should be referred to what it is (ablated tissue), not as scar since no scar has had time to develop.

Similarly, the manuscript should be revised to say potentially deliverable via dual lumen catheter and not described as a given since the authors have still not provided data showing this is feasible without clogging the tip of the catheter, which is not trivial. This reviewer appreciates this is in development, but it is not yet achieved and should not be stated as fact in the manuscript. Given that S31 just shows a prototype without delivering the material and therefore does not address the original concern, I would suggest this be removed from this manuscript, so it does not interfere with publishing a separate manuscript on the catheter once it is fully developed.

Other limitations have been adequately addressed with the revision.

Reviewer #3 (Remarks to the Author):

The authors have nicely addressed the comments. This work is suitable for publication in Nature Communications.

RESPONSE TO REVIEWERS:

The authors would like to thank the reviewers for their comments and enthusiasm for the innovative nature of this approach. We appreciate that the editorial team agreed that long-term validation was not necessary for this manuscript and have included detailed responses to individual reviewer comments below as well as a more detailed discussion of the evidence that supports the translational potential of this technology in the revised manuscript.

REVIEWER #1

The novelty of the approach with the initial experiments are interesting. The authors responded to earlier comments recognizing the need for additional experiments and development as well as longer-term follow-up.

REVIEWER #2

Given the journal transfer, I do not think additional data is required. However, text changes are needed per previous comments to more accurately reflect the data presented in the manuscript:

Comment 1: *The added limitation at the end of the manuscript is good, but all mention of testing the hydrogel in scarred myocardium or labelling ablated tissue as “scar” needs to be removed. The tissue was acutely evaluated after ablation. It should be referred to what it is (ablated tissue), not as scar since no scar has had time to develop.*

Response: We have removed the remaining reference to “scar” from the SI figures and text and referred to it as ablation lesion per standard terminology for this model.

Comment 2: *Similarly, the manuscript should be revised to say potentially deliverable via dual lumen catheter and not described as a given since the authors have still not provided data showing this is feasible without clogging the tip of the catheter, which is not trivial. This reviewer appreciates this is in development, but it is not yet achieved and should not be stated as fact in the manuscript. Given that S31 just shows a prototype without delivering the material and therefore does not address the original concern, I would suggest this be removed from this manuscript, so it does not interfere with publishing a separate manuscript on the catheter once it is fully developed.*

Response: We have revised the text accordingly to indicate that this material design allows for the potential of endovascular delivery via a dual lumen catheter. We retained the supplemental figure and modified the details to ensure that it does not limit future publication.

“Clinical experience together with the current data supports the potential of transvenous catheter-based delivery of the hydrogel electrode to the AIV through the subclavian vein or internal jugular.”

REVIEWER #3

The authors have nicely addressed the comments. This work is suitable for publication in Nature Communications.